# Preoperative Immunocite-Derived Ratios Predict Surgical Complications Better when Artificial Neural Networks Are Used for Analysis—A Pilot Comparative Study

**DOI:** 10.3390/jpm13010101

**Published:** 2023-01-01

**Authors:** Stefan Patrascu, Georgiana-Maria Cotofana-Graure, Valeriu Surlin, George Mitroi, Mircea-Sebastian Serbanescu, Cristiana Geormaneanu, Ionela Rotaru, Ana-Maria Patrascu, Costel Marian Ionascu, Sergiu Cazacu, Victor Dan Eugen Strambu, Radu Petru

**Affiliations:** 1Sixth Department of Surgery, University of Medicine and Pharmacy of Craiova, 200349 Craiova, Romania; 2Department of Medical Informatics and Statistics, University of Medicine and Pharmacy of Craiova, 200349 Craiova, Romania; 3Emergency Medicine Department, University of Medicine and Pharmacy of Craiova, 200342 Craiova, Romania; 4Hematology Department, University of Medicine and Pharmacy of Craiova, 200349 Craiova, Romania; 5Department of Statistics and IT, University of Craiova, 200585 Craiova, Romania; 6Department of Gastroenterology, University of Medicine and Pharmacy of Craiova, 200349 Craiova, Romania; 7Department of Surgery, “Carol Davila” Clinical University Hospital, 010731 Bucharest, Romania

**Keywords:** artificial neural network, biomarkers, inflammation, prognostic factors

## Abstract

We aimed to comparatively assess the prognostic preoperative value of the main peripheral blood components and their ratios—the systemic immune-inflammation index (SII), neutrophil-to-lymphocyte ratio (NLR), lymphocyte-to-monocyte ratio (LMR), and platelet-to-lymphocyte ratio (PLR)—to the use of artificial-neural-network analysis in determining undesired postoperative outcomes in colorectal cancer patients. Our retrospective study included 281 patients undergoing elective radical surgery for colorectal cancer in the last seven years. The preoperative values of SII, NLR, LMR, and PLR were analyzed in relation to postoperative complications, with a special emphasis on their ability to accurately predict the occurrence of anastomotic leak. A feed-forward fully connected multilayer perceptron network (MLP) was trained and tested alongside conventional statistical tools to assess the predictive value of the abovementioned blood markers in terms of sensitivity and specificity. Statistically significant differences and moderate correlation levels were observed for SII and NLR in predicting the anastomotic leak rate and degree of postoperative complications. No correlations were found between the LMR and PLR or the abovementioned outcomes. The MLP network analysis showed superior prediction value in terms of both sensitivity (0.78 ± 0.07; 0.74 ± 0.04; 0.71 ± 0.13) and specificity (0.81 ± 0.11; 0.69 ± 0.03; 0.9 ± 0.04) for all the given tasks. Preoperative SII and NLR appear to be modest prognostic factors for anastomotic leakage and overall morbidity. Using an artificial neural network offers superior prognostic results in the preoperative risk assessment for overall morbidity and anastomotic leak rate.

## 1. Introduction

There is growing evidence of a relationship between systemic inflammation and colorectal cancer (CRC), most notably between the inflammatory response and prognosis, overall survival, and disease-free survival [1,2,3]. One area of concern in oncologic surgery is postoperative complications, with particular emphasis on anastomotic leaks, which are positively correlated with worse prognosis and higher mortality rates [4]. Although surgical intervention elicits a temporary and limited local and general inflammatory response, preoperative local and systemic inflammation plays an equally important role in the postoperative course and may interfere with local healing processes.

While local inflammation can be partially assessed based on the level of peritumoral inflammatory infiltrate, systemic inflammation uses markers such as acute-phase proteins, circulating immune cells, and various blood-cell-derived formulas, including the systemic immune-inflammation index (SII), neutrophil-to-lymphocyte ratio (NLR), lymphocyte-to-monocyte ratio (LMR), and platelet-to-lymphocyte ratio (PLR) [5]. Despite being extensively investigated for their prognostic role in postoperative settings, little effort has been invested into analyzing the role of the four blood formulas as markers for complications in the preoperative period. This aspect should be addressed more thoroughly since SII, NLR, LMR, and PLM are plausible indicators of preoperative immune and metabolic status, and their variation may reflect an altered stress response, leading to poor postoperative outcomes and delayed postoperative recovery [6,7,8,9].

On the other hand, artificial neural network (ANN) is a modern machine-learning data-analysis tool based on the study of certain functions of biological nervous systems, involving numerous highly interconnected processing units working together. Although machine-learning techniques have been successfully implemented in various domains, including healthcare, ANN has only been sporadically used as a predictive instrument for colorectal cancer surgery [10,11,12]. Most often, ANNs proved their efficiency in colorectal cancer screening, diagnosis, survival assessment, and even as prediction tools for lymphovascular invasion in CRC [13,14,15,16].

More recently, ANNs found their use as predictors for treatment response following adjuvant therapy or as intraoperative detection instruments for CRC [17,18]. However, rather inexplicably, no specific assessment of their role in preoperatively detecting cases at risk of untoward events postoperative has been carried out to date. As a simple, reliable, and inexpensive preoperative prognostic instrument for surgical complications, ANN may prove essential in reducing the morbidity and mortality associated with this type of surgery. Moreover, it would allow the enhancement of surgical planning with regards the optimal timing and selection of the appropriate surgical procedure, thus performing a preemptive rather than a diagnostic role.

We aimed to assess the predictive power for surgical complications of ANN over traditional statistical analysis for colorectal cancer surgery based on the main preoperative peripheral-blood components and their ratios.

## 2. Materials and Methods

### 2.1. Study Population and Design

We retrospectively included all patients diagnosed with adenocarcinoma of the colon and rectum for which they underwent elective operation between 1 January 2015 and 31 December 2021 in a tertiary referral surgical center in Southwest Romania. 

For inclusion, each case had to fulfill the following criteria: (i) elective admission and surgery for colorectal cancer; (ii) histopathology diagnosis of adenocarcinoma; (iii) staging performed using contrast-enhanced CT of the thorax, abdomen, and pelvis, and (iv) treatment with radical surgery. Patients with chemotherapy or steroid medication and those with additional infectious, autoimmune or septic conditions during admission were excluded. Similarly, cases where definitive histopathologic results indicated positive resection margins were excluded from this study.

### 2.2. Perioperative Management

A multidisciplinary tumor board approved the therapeutic management of each patient. Patients with distant metastases were first submitted to first-line chemotherapy. 

All patients received mechanical bowel preparation using 4 L of polyethylene glycol starting two days prior to the surgical procedure and received three oral doses of 500 mg metronidazole daily after finishing the mechanical preparation. Patients under oral anticoagulants due to specific comorbidities were switched to receiving low-molecular-weight heparins (LMWH) 96 h before the surgical procedure, with the last administration of LMWH being 12 h before surgery and resuming 12 h after surgery. Prophylactic antibiotics were administered with induced anesthesia consisting of a second-generation cephalosporin, in accordance with our institution’s perioperative management protocols. After patient positioning, temporary nasogastric tube (NGT) and indwelling catheter (IDC) were used in each case. NGT was removed upon the completion of the procedure, while IDC was taken off at 24 h after surgery.

Surgery was performed either by open or by laparoscopic approach using mechanical anastomoses in most cases. The surgeon and anesthetic team’s approach and surgical technique were decided according to patients’ comorbidities, tumor size, previous abdominal surgery, and anticipated difficulties. Before bowel transection was performed, the evaluation of vascularization was performed by both mesenteric pulsation and serosal color assessment.

At the end of each procedure, the anastomosis integrity was inspected visually and was verified by air-leak test and methylene-blue-stained saline tests. In selected cases, near -nfrared fluorescence angiography with Indo-cyanine green was used to assess tissue perfusion at perianastomotic sites. Any leakage detected during intraoperative anastomotic assessment was managed either by takedown of the anastomosis with subsequent reconstruction or by anastomotic repair, with or without proximal diversion ileostomy. Passive abdominal drainage was used in every case, with drains removed when the total fluid output was 50 mL or less over 24 h or when patients had bowel movements. Patients were allowed liquids at 24 h after surgery and gradually proceeded to normal diet after no less than 48 h.

All patients were routinely followed at 2, 4, and 6 weeks postoperatively in the ambulatory care setting. They were advised to contact the designated surgeon and contact our institution for detailed assessment if they experienced any significant postoperative issues.

### 2.3. Data Extraction

Complications after surgery were classified into three categories based on the Clavien–Dindo scale: a mild-complication group, corresponding to Clavien–Dindo grades 1–2; moderately severe, corresponding to Clavien–Dindo grades 3–4; and fatal complication (Clavien–Dindo grade 5) [19]. Anastomotic leakage was defined according to the recommendations of the International Study Group of Rectal Cancer [20]. 

Anastomotic leak was diagnosed based on radiologic examination using contrast medium, leakage on the abdominal drains, evidence of perianastomotic abscess, and peritonitis. 

All data concerning age, sex, comorbidities, CBC, postoperative complications, and pathology reports were retrieved from our institution’s electronic data system. Patients had their complete blood count determined 24 to 48 h before surgery. Based on these values, NLR, LMR, and PLR were calculated and compared between various subgroups. Correlations between the levels of inflammatory markers and the occurrence of anastomotic fistula or the degree of complications were systematically searched and analyzed. 

### 2.4. Statistical Analysis

Data analysis was performed using SPSS 20.0 software. Categorical variables were represented as absolute and percentage values. For quantitative variables, Student’s *t*-testing and one-way ANOVA were performed to determine between-group differences in terms of anastomotic leak and the severity of postoperative complications. The result was considered statistically significant if *p* < 0.05, corresponding to a 95% confidence interval. Pearson correlation coefficient was used to determine correlations between NLR, PLR, and LMR values and the occurrence of anastomotic leakage and overall surgical complications. The optimal threshold value was established after analyzing the receiver operating characteristic (ROC) curves and area under the curve (AUC) values based on Youden’s index to optimize the specificity and sensitivity of the preoperative inflammatory response markers in predicting the event. Logistic regression analysis was used to assess the likelihood of a specific event based on heterogenous factors.

### 2.5. Artificial Neural Network

The analyzed variables of <hemoglobin>, <SII>, <NLR>, <LMR>, <PLR>, and <serum protein> were used to predict three clinical aspects of interest (tasks): anastomotic leak (y/*n*), complication (y/*n*), and complication groups (0–3) based on the abovementioned classification. The dataset contained no missing values.

The database was balanced via the up-sampling of the minority classes, such that the class distribution was similar. Overfitting the majority class was repressed with this operation.

The balanced dataset was split into 80% training, 10% validation, and 10% testing. A feed-forward fully connected multilayer perceptron (MLP)-analysis-classification network was trained for each task. The hidden layer contained 90 neurons for each network [21]. Several hidden-layer neurons were empirically set, increasing the number of neurons and, thus, the network accuracy (ACC) until stable performance was achieved. Due to their stochastic characteristics, the networks were run 100 times in a cross-validation sequence to achieve reliable results. The ACC, sensitivity, and specificity were computed for each run on the test data, and the results are presented as mean and standard deviation (SD). The algorithm was implemented using MATLAB (MathWorks, Natick, MA, USA). The network structures are presented in Figure 1 and Figure 2. 

For summarizing the performance of the MLP-classification model, a two-class confusion matrix was used to display the predicted state for each pattern compared with the actually occurring value of the output unit for the inquired variable. The results were characterized by accuracy, error rate (ERR), sensitivity, specificity, precision and false-positive rate (FPR). The F score, Matthews correlation coefficient (MCC), and Cohen’s Kappa coefficient (K) measurements were used in the overall assessment of the classification.

## 3. Results

### 3.1. Patient Characteristics

We identified 415 cases of colon and rectal cancer admitted to our institution between 2015 and 2021, 308 of which were submitted to elective surgery. Twenty-seven patients were excluded from the study due to recent corticoid-therapy use or after being diagnosed with a different subtype of cancer (other than adenocarcinoma). In 72 cases of right-colon cancer, right hemicolectomy with ileocolonic anastomosis was performed. Fourteen cases were located in the transverse colon and treated by either transverse-colon resection or extended right hemicolectomy. In twenty-four cases, the tumor was located on the splenic flexure or in the descending colon, for which a left hemicolectomy was performed. Sigmoidectomy was the procedure of choice for 67 patients with sigmoid cancer. A rectosigmoidectomy was performed for patients with tumors located on the rectosigmoid junction. For the 85 patients with rectal cancer, either low anterior resection (59 cases) or abdominoperineal excision (26 cases) were performed, whereas in two cases, a total rectocolectomy was performed due to associated colonic polyposis. The total number of procedures that involved an end colostomy was 74.

The ages of the patients were in the range of 27–96 years, with an average of 68.62 years (95% CI: 67.4 to 69.84) and a sex ratio (M/F) of 1.35. The patients’ preoperative comorbidities were mainly cardiovascular diseases (53.7%), diabetes mellitus (13.5%), and obesity (10.3%). The histopathologic assessment showed that the majority of the tumors were either T3 (49.5%) or T4 (32.4%), with T2 tumors accounting for only 18% of the cases. The N1 and N2 were encountered in 35.2 and 26% of cases, and metastases were diagnosed in 18.1% of cases (Table 1).

Anastomotic leakage was encountered in 24 cases (8.5%), with most of the cases occurring after low anterior resections or ileocolic anastomosis. The main biological parameters analyzed are presented in Table 2.

No significant difference in the values of the main preoperative blood cell variables or protein levels was found between patients with anastomotic leaks and those with no adverse anastomotic outcomes. The main variable in the two groups that displayed statistical differences was the hemoglobin level (*p* = 0.022). However, when the cases with ostomy creation (both protective stoma and end colostomy) were excluded, statistical differences in the mean preoperative SII and NLR levels were found between the anastomotic leak group and the group without anastomotic complications. Similarly, the preoperative NLR values and, to a lesser extent, the SII values, were correlated with the postoperative development of anastomotic leakage (NLR: *r* = 0.42, α = 0.001; SII: r = 0.251, α = 0.001). Despite having statistically different values (*p* = 0.04), no correlations were observed between the preoperative LMR and the anastomotic leaks (Table 3).

For the overall postoperative morbidity analysis, statistical differences in SII, NLR, and PLR were found between the patients with and without postoperative complications, with medium-level correlations observed for SII and NLR (SII: r = 0.33, *p* < 0.001; NLR: r = 0.4, *p* < 0.001) (Table 4).

In the case of anastomotic leakage, after generating the ROC curve for a cut-off value of 2.998 for NLR, the sensitivity was 0.682, and the specificity was 0.561. The optimal cut-off value for SII as a predictor for anastomotic leakage was 793, with a sensitivity of 63% and specificity of 53%. The cut-off value of NLR and SII for maximum sensitivity and specificity for postoperative morbidity prediction were 3.26 (sensitivity of 0.736, specificity of 0.624) and 933 (sensitivity of 0.667, specificity of 0.613), respectively. Moreover, in the case of NLR and SII as predictors of leaks, ROC-curve analysis indicated that the AUCs were 0.711 and 0.622, respectively. Regarding the predictive value of NLR and SII for postoperative complications, the ROC-curve analysis indicated that the AUCs were 0.774 and 0.702, respectively.

On the multivariate regression analysis, NLR was the only variable to exert a significant effect on the postoperative leak rate (OR 3.159, 95% CI 1.328–7.517, *p* = 0.009) (Table 5). Concerning the overall postoperative complication rate, the multivariate analysis indicated the thrombocyte level as the only factor exerting influence on the surgical morbidity (OR 0.990, 95% CI 0.982–0.999, *p* = 0.023) (Table 6).

### 3.2. MLP Neural Networks

For the first task, <anastomosis leak (y/*n*)>, the resulting values for accuracy, sensitivity, and specificity were 0.79 ± 0.08, 0.78 ± 0.07, and 0.81 ± 0.11, respectively. For the second task, <complication (y/*n*)>, the resulting values for accuracy, sensitivity, and specificity were 0.71 ± 0.03, 0.74 ± 0.04, and 0.69 ± 0.03, respectively. For the third task, <complication class>, the resulting values for accuracy, sensitivity, and specificity were 0.70 ± 0.13, 0.71 ± 0.13, and 0.9 ± 0.04, respectively.

The confusion matrix used for the assessment of the performance of the <leak> task showed an accuracy of 0.9202, ERR of 0.0798, sensitivity of 0.8624, specificity and precision of 1, and FPR of 0. The F1 score was 0.9261, while the MCC was 0.8514 and the K was 0.8405. The confusion matrix used for <complication> indicated a lower performance of the system in terms of accuracy, 0.7835; ERR, 0.2165; sensitivity, 0.8090; specificity, 0.7619; precision, 0.7423; and FPR, 0.2381. In this instance, the F1 score was 0.7742, with a MCC of 0.5689 and K of 0.5670. The results of the best-performing networks are presented in Table 7.

Thus, the ANN’s predictive power exceeded the conventional statistics for immunocyte-derived ratios in terms of sensitivity and specificity.

## 4. Discussion

The results indicate that certain preoperative immunocyte-derived ratios share a moderate degree of correlation with the risk of untoward postoperative events in colorectal surgery, especially anastomotic leaks. The use of machine-learning algorithms, such as a feed-forward fully connected MLP analysis, can significantly improve the prediction power for these types of events.

Inflammation has long been suspected as being one of the fundamental mechanisms involved in the oncogenesis and pathophysiology of cancer [22,23]. Although the overall relationship between chronic inflammation and neoplasia is unquestionable for inflammatory conditions such as Crohn’s disease and ulcerative colitis, there is probably a certain alteration in the inflammatory response for all colorectal cancer subtypes, regardless of the etiology [24,25,26].

A multitude of molecules and scores, such as cytokines, platelet transcriptome, growth factors, intestinal damage markers (such as intestinal-fatty-acid-binding protein, liver-fatty-acid-binding protein, and calprotectin), and even peritoneal-drain-fluid analysis, have been investigated as markers of poor prognosis in oncologic patients with rather inconclusive results [27,28,29]. However, most markers are difficult to implement in current clinical practice and offer poor predictive accuracy, albeit with a potential improvement when combined analysis is used [30]. One of the easiest and most widely available solutions for this topic is the careful assessment of the blood count and relative variation of its different subsets of elements, such as NLR, LMR, PLR, and SII [31,32,33].

A more modern solution is the use of ANN and machine-learning algorithms, which are increasingly attractive solutions for outcome prediction in a variety of fields, including clinical medicine, due to their impressive learning and improvement potential. 

Our study provides a more comprehensive analysis of the systemic inflammatory changes occurring in the preoperative period for colorectal cancer patients based on four widely accessible markers. NLR, LMR, and PLR have been tested for their prognostic value in a wide spectrum of diseases, such as sepsis, renal, metabolic, infectious, or cardiovascular pathology [34,35,36,37]. In a limited number of studies, SII was successfully used for predicting oncologic outcomes for lung, colorectal, and renal cancer [38,39,40]. The role of NLR, LMR, and PLR has been analyzed in the prediction of colorectal cancer postoperative morbidity in only a few studies [41,42,43], whereas no such study is available for SII. Moreover, most of the abovementioned studies analyze the postoperative values of these indices, allowing the early detection of untoward postoperative events, but doing little to prevent them by tackling the conditions leading to the inflammatory status in the preoperative setting.

On the other hand, artificial intelligence and machine-learning-driven models can potentially improve the planning of suitable surgical protocols by reliably discerning individuals who are more prone to experiencing postoperative complications while also dealing with the limitations of conventional statistics-based risk-assessment solutions [10].

Based on these premises, our objective for this study was to evaluate potential correlations between the immunocyte-derived ratios in the preoperative setting and adverse events following colorectal surgery. Since most of these adverse events are related to abdominal sepsis, selecting these markers was a reasonable option, especially in the context of the altered inflammatory response associated with colorectal cancer. By comparing these results to those of the ANN, we attempted to overcome some of the drawbacks of conventional statistics, as neural-network models have better classification accuracy and improve the assessment of correlated independent variables and nonlinear relations [44]. Moreover, although ANN was used in certain fields of digestive surgery for survival prediction and surgical complications, the use of ANN for the prediction of adverse postoperative events in colorectal surgery remains limited [45,46,47].

Our study found significant differences in the NLR and SII values between patients without anastomotic leakage and those with postoperative leaks; the correlation level pointed toward a potentially preexisting alteration in the immune response that may have been a risk factor for anastomotic leakage. Similarly, we observed statistical differences and positive correlations when assessing the variation in the NLR and SII values in patients with postoperative complications, unlike PLR and LMR. However, in the multivariate analysis, neither SII nor NLR were independent prognostic factors for overall surgical complications.

The results of this pilot study indicate that trained neural networks provide results with superior sensitivity and specificity for assessing postoperative outcomes in colorectal cancer surgery than conventional statistics-based risk evaluations. As more data and, possibly, more variables are added to the database, one can expect improvement in the predictive value of the ANN model.

Our study has several limitations, such as the low number of patients in each sub-group and the monocentric and retrospective nature of the analysis, potentially limiting the relevance of its findings. However, this study can serve as a basis for further research regarding the role of ANN and blood-cell-ratio analysis as preoperative predictors for negative outcomes following colorectal surgery.

## 5. Conclusions

The findings of our study indicate that, unlike LMR and PLR, preoperative NLR and, to a lesser extent, SII, may be used as useful preoperative predictors for anastomotic leak and postoperative morbidity in colorectal cancer surgery. The integration of machine learning and ANNs in the predictive algorithm of postoperative complications in elective colorectal cancer surgery provides superior results compared with traditional statistics.

## Figures and Tables

**Figure 1 jpm-13-00101-f001:**
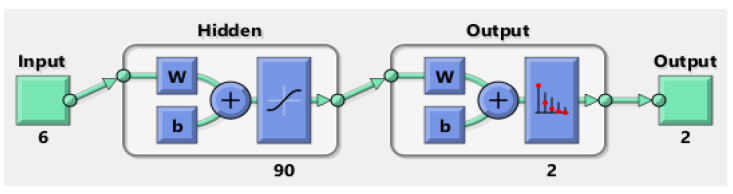
Network structure for <leak> and <complication (y/*n*)>.

**Figure 2 jpm-13-00101-f002:**
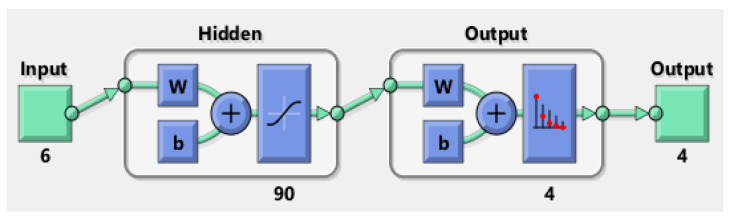
Network structure for <complication class>.

**Table 1 jpm-13-00101-t001:** Clinical and demographic characteristics of the enrolled patients.

Factor	Subtype/Measurement	*n* (281)	Percentage (%)
Age	Mean ± SD	68.70 ± 10.55	
	Range (y)	27–96	
Sex	M	160	56.9%
	F	121	43.1%
Localization	Right colon	72	25.62%
	Transverse colon	14	4.98%
	Descendent colon	24	8.54%
	Sigmoid colon	67	23.84%
	Rectosigmoid junction Rectum	23 81	8.18% 28.82%
Comorbidities	Cardiovascular diseases	151	53.7%
	Diabetes mellitus	38	13.5%
	Obesity	29	10.3%
Staging	T_4_	91	32.4%
	T_3_	139	49.5%
	T_2_	42	14.9%
	T_1_	9	3.2%
	N_0_	109	38.8%
	N_1_	99	35.2%
	N_2_	73	26.0%
	M_1_	51	18.1%
Postoperative complications		87	30.9%
	Anastomotic leakage	24	8.5%
	Wound	31	11.0%
	Sepsis	12	4.3%
	Cardiovascular event Other	17 24	6.0% 8.5%
Complication grade (Clavien–Dindo grade)	Mild complications (1–II)	39	13.8%

	Moderately severe complications (III–IV)	27	9.6%

	Severe complications (V)	21	7.47%

**Table 2 jpm-13-00101-t002:** Preoperative biological characteristics of the enrolled patients.

Variable	Avg ± SD	95% CI
Hb	10.92 ± 2.44	10.64–11.20
Platelet	283.62 ± 119.84	269.77–297.47
Neutrophils	6.276 ± 4.45	5.76–6.79
SII	1301.99 ± 1441.31	1132.73–1471.24
NLR	4.47 ± 4.09	3.78–4.98
LMR	3.45 ± 3.00	3.10–3.80
PLR	194.82 ± 131.73	179.60–210.05

Abbreviations: SII, systemic immune-inflammation index; NLR, neutrophil-to-lymphocyte ratio NLR; LMR, lymphocyte-to-monocyte ratio; PLR, platelet-to-lymphocyte ratio.

**Table 3 jpm-13-00101-t003:** Mean values of preoperative immunocyte-derived markers in patients with anastomotic leakage after colorectal cancer surgery.

	Mean ± SD	95% CI	*p* (*t*-Test)	*R* (Pearson)
SII				
Without AL (*n* = 164)	993.35 ± 878.94	857.82 to 1128.87	0.001	0.251
With AL (*n* = 22)	1913.19 ± 2368	863.17 to 2963.20		
NLR				
Without AL (*n* = 164)	3.17 ± 1.70	2.91 to 3.43	0.001	0.42
With AL (*n* = 22)	6.73 ± 5.54	4.27 to 9.19		
LMR				
Without AL (*n* = 164)	4.04 ± 3.48	3.51 to 4.58		0.06
With AL (*n* = 22)	2.50 ± 1.65	1.77 to 3.24	0.04	
PLR				
Without AL (*n* = 164)	177.32 ± 86.10	164.04 to 190.60		0.14
With AL (*n* = 22)	195.66 ± 125.97	139.80 to 251.51	0.37	

Abbreviations: AL, anastomotic leakage; SII, systemic immune-inflammation index; NLR, neutrophil-to-lymphocyte ratio NLR; LMR, lymphocyte-to-monocyte ratio; PLR, platelet-to-lymphocyte ratio.

**Table 4 jpm-13-00101-t004:** Variation in preoperative immunocyte-derived markers in relation to postoperative complications.

Grade of Complication	Avg ± St Dev	95% CI	*p* (ANOVA)	*r* (Pearson)
SII				
No complications (*n* = 194)	897.99 ± 571.60	817.05 to 978.93	0.001	0.341
Mild complications (*n* = 39)	2331.28 ± 2064.51	1662.04 to 3000.52		
Moderately severe complications (*n* = 27)	2224.32 ± 2427.79	1263.92 to 3184.72		
Severe complications (*n* = 21)	1936.72 ± 2224.67	924.06 to 2949.38		
NLR				
No complications (*n* = 194)	3.15 ± 1.58	2.92 to 3.37	0.001	0.412
Mild complications (*n* = 39)	7.57 ± 4.62	6.08 to 9.07		
Moderately severe complications (*n* = 27)	7.66 ± 6.55	5.07 to 10.25		
Severe complications (*n* = 21)	6.69 ± 6.90	3.55 to 9.84		
LMR				
No complications (*n* = 194)	4.50 ± 9.71	3.13 to 5.88	0.107	0.096
Mild complications (*n* = 39)	2.09 ± 1.15	1.71 to 2.46		
Moderately severe complications (*n* = 27)	2.61 ± 1.59	1.98 to 3.25		
Severe complications (*n* = 21)	2.70 ± 1.84	1.86 to 3.55		
PLR				
No complications (*n* = 194)	175.67 ± 81.30	164.15 to 187.18	0.002	0.188
Mild complications (*n* = 39)	246.81 ± 139.02	201.75 to 291.88		
Moderately severe complications (*n* = 27)	211.09 ± 133.46	158.29 to 263.89		
Severe complications (*n* = 21)	250.46 ± 315.77	106.73 to 394.20		

Abbreviations: SII, systemic immune-inflammation index; NLR, neutrophil-to-lymphocyte ratio NLR; LMR, lymphocyte-to-monocyte ratio; PLR, platelet-to-lymphocyte ratio.

**Table 5 jpm-13-00101-t005:** Factors influencing postoperative leak rate in multivariate regression analysis.

Variable	Odds Ratio (OR)	95% CI		*p* Value
Age	1.067	0.989	1.151	0.093
Obesity	76.149	0.096	60,360.412	0.203
Diabetes	0.197	0.021	1.836	0.154
Local tumor extension (T)				0.652
T 1	6.744	0.010	4477.042	0.565
T 2	2.245	0.207	24.303	0.506
T 3	2.958	0.531	16.483	0.216
Lymph-node extension (*n*)				0.233
*n* 1	0.618	0.051	7.462	0.705
*n* 2	2.702	0.306	23.849	0.371
Metastasis (M)	0.400	0.072	2.226	0.296
Hb	1.531	1.043	2.247	0.030
Thrombocytes	1.002	0.989	1.015	0.806
NLR	3.159	1.328	7.517	0.009
SII	0.998	0.996	1.001	0.136
PLR	0.999	0.986	1.012	0.857
LMR	1.040	0.790	1.368	0.779
Proteins	0.691	0.280	1.707	0.423

Abbreviations: SII, systemic immune-inflammation index; NLR, neutrophil-to-lymphocyte ratio NLR; LMR, lymphocyte-to-monocyte ratio; PLR, platelet-to-lymphocyte ratio.

**Table 6 jpm-13-00101-t006:** Factors influencing overall surgical-complications rate in multivariate regression analysis.

Variable	Odds Ratio (OR)	95% CI		*p* Value
Age	1.015	0.977	1.054	0.452
Sex	2.020	0.891	4.579	0.092
Obesity	1.666	0.348	7.983	0.523
Diabetes	0.960	0.300	3.071	0.945
Local tumor extension (T)				0.916
T 1	0.624	0.031	12.734	0.759
T 2	0.782	0.212	2.884	0.712
T 3	0.726	0.296	1.783	0.485
Lymph-node extension (*n*)				0.476
*n* 1	0.517	0.174	1.539	0.236
*n* 2	0.618	0.222	1.717	0.356
Metastasis (M)	1.113	0.415	2.983	0.832
Hb	1.020	0.830	1.252	0.852
Thrombocytes	0.990	0.982	0.999	0.023
NLR	1.104	0.628	1.626	0.648
SII	1.002	0.723	1.686	0.051
PLR	0.999	1.000	1.003	0.503
LMR	0.986	0.995	1.003	0.898
Proteins	1.010	0.799	1.217	0.966

Abbreviations: SII, systemic immune-inflammation index; NLR, neutrophil-to-lymphocyte ratio NLR; LMR, lymphocyte-to-monocyte ratio; PLR, platelet-to-lymphocyte ratio.

**Table 7 jpm-13-00101-t007:** The confusion matrix of predicted and true postoperative outcomes after colorectal cancer surgery in the MLP network.

		Predicted Classes			
		Anastomotic Leakage		Complication	
		Yes	No	Yes	No
True classes	Yes	257	41	144	34
	No	0	216	50	160

## Data Availability

Data available on request.

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
