# Peer review of "Preoperative Immunocite-Derived Ratios Predict Surgical Complications Better when Artificial Neural Networks Are Used for Analysis—A Pilot Comparative Study"

_jpm, 2023, doi:10.3390/jpm13010101_

Round 1

Reviewer 1 Report

Dear Authors, 

This paper is structured in a simple way, which is an advantage. Basic introduction, a few facts about medical treatment and assumptions for study. Study is clearly presented and most of the statements come from data/results. Thus I have no doubts that this paper is needed and highly welcome in the scientific world. 

Nonetheless, I found a few mistakes, some parts of the manuscript in my humble opinion should be modified/revised:

  1. It is strange and suspicious: all authors provided gmail/yahoo mailboxes. Any professional mailbox, even for the corresponding author.

  2. Abstract is sectioned in quite an unordinary way. I think it should be a plain piece of the text.

  3. Literature [3] is quite an old source, and strong statements are based on that. Any newer papers which could prove that this is still valid knowledge?

  4. Line 64 - I have a kind of allergy when authors write about similarities about ANN and biological nervous tissue. In fact, only what you can write is that McCulloch-Pitts neuron is inspired by a nervous cell. That is it, nothing more. 

  5. There is very limited state of the art. You just cited a bunch of papers [10-12] without giving any wider context. Even in discussion you try to compare your results with others while it should be done in state of the art. 

  6. In the last sentence of introduction you emphasize your hypothesis, which is ok for me, and throughout the paper you are trying to prove it, but my recommendation is to highlight it as a single indent.

  7. For table 5 and 6, the fourth column header is missing.

  8. Have you tried any reduction dimensionality technique, for instance PCA? This may help to exclude data which disperse your correlation in data.

  9. Since one third of reading this paper I had an idea of creating a simple workflow for a covered topic and in conclusion I found the same observation. In other words, I think it might be beneficial if you provide a simple workflow or even a simple tool for other medical centers, where you describe how to  proceed with such data. In a perfect world you can even establish a kind of webpage where such cases will be reported by other researchers and added to  the database, because your database is relatively small.

Author Response

Dear Sir,

We are grateful for the insightful comments on our paper. We were able to incorporate the changes to address most of the observations and suggestions you provided. With your permission, we will discuss them separately in the lines below:

1. It is strange and suspicious: all authors provided gmail/yahoo mailboxes. Any professional mailbox, even for the corresponding author.

    • Thank you for raising this issue. As our Institutions do not require the use of professional mailboxes for Journal correspondence, personal e-mail was used for better feed-back. Moreover, for those authors with UMF affiliation, a recent site update (https://www.umfcv.ro/) led to some login issues during the last two months. The problem is now solved and, if deemed necessary, institutional professional mailbox can be provided.

2. Abstract is sectioned in quite an unordinary way. I think it should be a plain piece of the text.

  • The abstract is now structured as a plain piece of text.

3. Literature [3] is quite an old source, and strong statements are based on that. Any newer papers which could prove that this is still valid knowledge?

  • Thank you very much for pointing this out. I believe you were making refference about this source: Refference “ Wako, G.; Teshome, H.; Abebe, E. Colorectal Anastomosis Leak: Rate , Risk Factors and Outcome in a Tertiary Teaching Hospital, Addis Ababa Ethiopia, a Five Year Retrospective Study. Ethiop. J. Health Sci. 1970, 29, 767–774. https://doi.org/10.4314/ejhs.v29i6.14.” , which was replaced with
  • “Holmgren, K.; Jonsson, P.; Lundin, C.; Matthiessen, P.; Rutegård, J.; Sund; M.; Rutegård, M. Preoperative biomarkers related to inflammation may identify high-risk anastomoses in colorectal cancer surgery: explorative study. BJS open, 2022, 6(3), zrac072. https://doi.org/10.1093/bjsopen/zrac072.”

4. Line 64 - I have a kind of allergy when authors write about similarities about ANN and biological nervous tissue. In fact, only what you can write is that McCulloch-Pitts neuron is inspired by a nervous cell. That is it, nothing more. 

  • We completely agree, the similarities between biological nervous tissue and ANN are crudely overstated. The text (lines 64 and 65) may have been confusing, therefore we made some modifications for clarity.

5. There is very limited state of the art. You just cited a bunch of papers [10-12] without giving any wider context. Even in discussion you try to compare your results with others while it should be done in state of the art. 

  • Revised accordingly. We detailed the current context for this research by providing additional data and references.

6. In the last sentence of introduction you emphasize your hypothesis, which is ok for me, and throughout the paper you are trying to prove it, but my recommendation is to highlight it as a single indent.

  • Revised accordingly. Single indent is now used to highlight our hypothesis

7. For table 5 and 6, the fourth column header is missing.

  • The issue was fixed, the column header was in fact marking the confidence interval and was not appropriately centered. The changes are made and are highlighted (red).

8. Have you tried any reduction dimensionality technique, for instance PCA? This may help to exclude data which disperse your correlation in data.

  • Thank you for your kind suggestion. PCA would probably bring better results if we had a larger number of variables. However, in our humble opinion, its use would make an unfair comparison with the classical statistics.

9. Since one third of reading this paper I had an idea of creating a simple workflow for a covered topic and in conclusion I found the same observation. In other words, I think it might be beneficial if you provide a simple workflow or even a simple tool for other medical centers, where you describe how to  proceed with such data. In a perfect world you can even establish a kind of webpage where such cases will be reported by other researchers and added to  the database, because your database is relatively small.

  • Thank you for your suggestion. Indeed, as this pilot study has a relatively small database, the creation of a workflow would undoubtedly prove beneficial. We have created a google form where anyone could add their own data. The dataset contains the variables used in our research and several identification variables. The form could be found following this link https://docs.google.com/forms/d/e/1FAIpQLSfT_oa4yQdb3GbIDqnAJsneKn0RGXNCDqJTOoRdSFCVRR60dg/viewform

This form may be used in the current manuscript or can be expanded for further research.

Respectfully Yours,

On behalf of the Authors

Stefan PATRASCU

Reviewer 2 Report

In my comments, I will focus on the use of Artificial Neural Networks which is the focus of the article.

My understanding is that the data used by the models consisted of 281 records. Each record included the values of the following predictors:  , , , , , and . It is unclear to me whether the data used to train and test the neural networks were the actual values of the variables or ratios computed out of the raw data (SII, NLR, LMR, PLR) or a combination of both. 

Three outcomes were considered: anastomotic leak (y/n), complication (y/n), and complication groups. Three distinct neural networks (two having the same structure) were trained and tested to predict the outcomes.

The article does not mention missing values. I assume there was no missing value but this should be explicitly stated.

The authors state (line 155) that data was split into training, validation, and test datasets. They also say (line 160) that 10-fold cross-validation was applied. These two sentences are not clear enough. A clarification of the roles of training, validation, and test datasets is required. In 10-fold cross-validation data is split into the train (90% of data) and test datasets (10% of data). How was the validation dataset built and what was its role?

The ANNs used in the article have two layers and 90 neurons. How did the authors arrive at this design? Were multiple different structures compared on the validation dataset and then the models were tested on the test dataset or a different approach was used?

Author Response

Dear Madame/Sir,

We would like to thank you for the careful and constructive analysis and remarks.

In the few lines below, we tried to address each one of them separately, making the appropriate changes in the body of the manuscript.

My understanding is that the data used by the models consisted of 281 records. Each record included the values of the following predictors:  , , , , , and . It is unclear to me whether the data used to train and test the neural networks were the actual values of the variables or ratios computed out of the raw data (SII, NLR, LMR, PLR) or a combination of both. 

  • Thank you very much for pointing this out. The ANN were trained and tested with the computed ratios (, , , ) which were based on the , , , , together with  and . The text may have been misleading and was modified for better clarity and eloquence.

Three outcomes were considered: anastomotic leak (y/n), complication (y/n), and complication groups. Three distinct neural networks (two having the same structure) were trained and tested to predict the outcomes.

The article does not mention missing values. I assume there was no missing value but this should be explicitly stated.

  • Thank you for your suggestion. There were no missing values; a specific mention is now introduced in text.

The authors state (line 155) that data was split into training, validation, and test datasets. They also say (line 160) that 10-fold cross-validation was applied. These two sentences are not clear enough. A clarification of the roles of training, validation, and test datasets is required. In 10-fold cross-validation data is split into the train (90% of data) and test datasets (10% of data). How was the validation dataset built and what was its role?

  • We used cross validation, and split the data as mentioned (80/10/10); on the training-validation sequence we used 80/10, while the reported (test) performance was made on the remaining 10%. We have adapted the text in order to clarify this aspect.

The ANNs used in the article have two layers and 90 neurons. How did the authors arrive at this design? Were multiple different structures compared on the validation dataset and then the models were tested on the test dataset or a different approach was used?

  • Thank you for your question. Several hidden layer neurons were empirically set, increasing the number of neurons and, thus, the network accuracy (ACC) until stable performance was achieved. No special algorithm was used, just a drop in ACC (non significant) between different iterations.

Respectfully Yours,

On behalf of the Authors

Stefan PATRASCU